# Gene Expression Pattern of *ESPL1*, *PTTG1* and *PTTG1IP* Can Potentially Predict Response to TKI First-Line Treatment of Patients with Newly Diagnosed CML

**DOI:** 10.3390/cancers15092652

**Published:** 2023-05-08

**Authors:** Eva Christiani, Nicole Naumann, Christel Weiss, Birgit Spiess, Helga Kleiner, Alice Fabarius, Wolf-Karsten Hofmann, Susanne Saussele, Wolfgang Seifarth

**Affiliations:** 1Department of Hematology and Oncology, Medical Faculty Mannheim, Heidelberg University, 68167 Mannheim, Germany; evalouise.christiani@gmail.com (E.C.); nicole.naumann@medma.uni-heidelberg.de (N.N.); birgit.spiess@medma.uni-heidelberg.de (B.S.); helga.kleiner@medma.uni-heidelberg.de (H.K.); alice.fabarius@medma.uni-heidelberg.de (A.F.); w.k.hofmann@medma.uni-heidelberg.de (W.-K.H.); susanne.saussele@medma.uni-heidelberg.de (S.S.); 2Department of Medical Statistics and Biomathematics, Medical Faculty Mannheim, Heidelberg University, 68167 Mannheim, Germany; christel.weiss@medma.uni-heidelberg.de

**Keywords:** *ESPL1*/Separase, *PTTG1*/Securin, *PTTG1IP*/Securin interacting protein, chronic myeloid leukemia (CML), *BCR::ABL1* expression, major molecular response (MMR), risk stratification at initial diagnosis, TKI first-line treatment

## Abstract

**Simple Summary:**

There is still a lack of reliable molecular predictors to achieve major molecular response (MMR, *BCR::ABL1* ≤ 0.1% IS) within the first year of treatment with tyrosine kinase inhibitors (TKI) in the therapeutic management of newly diagnosed chronic myeloid leukemia (CML). Employing a proprietary fluorogenic flow cytometry assay, we recently identified separase proteolytic activity as a potential marker of molecular response and *BCR::ABL1* positivity of CD34+ cells in TKI-treated CML patients. Here, we analyzed the expression and predictive value of *ESPL1*/Separase, *PTTG1*/Securin and *PTTG1IP*/Securin interacting protein transcript levels in white blood cells of CML patients (*n* = 97) at the time of diagnosis by means of qRT-PCR. We establish a novel distance (cut-off) score based on *ESPL1*, *PTTG1* and *PTTG1IP* gene expression levels that can serve as predictors of TKI non-response in about 10% of analyzed non-responding patients and may have potential benefit for the risk stratification of CML patients.

**Abstract:**

The achievement of major molecular response (MMR, *BCR::ABL1* ≤ 0.1% IS) within the first year of treatment with tyrosine kinase inhibitors (TKI) is a milestone in the therapeutic management of patients with newly diagnosed chronic myeloid leukemia (CML). We analyzed the predictive value of gene expression levels of *ESPL1*/Separase, *PTTG1*/Securin and *PTTG1IP*/Securin interacting protein for MMR achievement within 12 months. Relative expression levels (normalized to *GUSB*) of *ESPL1*, *PTTG1* and *PTTG1IP* in white blood cells of patients (responders *n* = 46, non-responders *n* = 51) at the time of diagnosis were comparatively analyzed by qRT-PCR. 3D scatter plot analysis combined with a distance analysis performed with respect to a commonly calculated centroid center resulted in a trend to larger distances for non-responders compared to the responder cohort (*p =* 0.0187). Logistic regression and analysis of maximum likelihood estimates revealed a positive correlation of distance (cut-off) with non-achieving MMR within 12 months (*p =* 0.0388, odds ratio 1.479, 95%CI: 1.020 to 2.143). Thus, 10% of the tested non-responders (cut-off ≥ 5.9) could have been predicted already at the time of diagnosis. Future scoring of *ESPL1*, *PTTG1* and *PTTG1IP* transcript levels may be a helpful tool in risk stratification of CML patients before initiation of TKI first = line treatment.

## 1. Introduction

### 1.1. Molecular Response in Treatment of Chronic Myeloid Leukemia

An improved therapy regimen employing selective tyrosine kinase inhibitors (TKI) directed at the abnormal *BCR::ABL1* tyrosine kinase led to durable cytogenetic (CyR) and molecular response (MR) in the majority of patients with chronic myeloid leukemia (CML), thus achieving an almost normal life expectancy [1,2]. TKI treatment leads to a reduction in malignant cells by several orders of magnitude, in some cases even below the level of detection using quantitative reverse transcriptase polymerase chain reaction (qRT-PCR), which is the gold standard method for patient follow-up [3,4,5].

The molecular response is assessed according to the International Scale (IS) as the ratio of *BCR::ABL1* transcripts to *ABL1* or beta-glucuronidase (*GUSB*) transcripts. It is reported as *BCR::ABL1%* on a log scale, where 1%, 0.1%, 0.01%, 0.0032%, and 0.001% correspond to a decrease of 2, 3, 4, 4.5, and 5 logs, respectively, below the standardized baseline that was established in the IRIS study [3,4,5]. A *BCR::ABL1* transcript level of ≤0.1% is defined as a major molecular response (MMR) or MR3. According to the current European LeukemiaNet (ELN), recommendations monitoring milestones of *BCR::ABL1* transcript levels at 3, 6, and 12 months determine whether the current treatment should be continued (optimal response), changed (failure/resistance), or carefully reconsidered, depending on patients’ characteristics, co-morbidities and tolerance (warning) [2,6,7]. Achieving an MMR (*BCR::ABL1* ≤ 0.1%) within 12 months predicts a CML-specific survival close to 100%, as disease progression is uncommon once this level of cytoreduction has been achieved. A change of treatment may be considered (warning) if MMR is not reached by 12 months [2,8].

There is still a search for robust diagnostic prediction markers for the desired fast and/or deep response to TKI first-line treatment. Such markers could improve patient management by helping to select patients at diagnosis that are a priori candidates for intensified treatment concepts, also justifying the potential risk of side effects, preventing over- or under-treatment and saving costs [9]. So far, response-related dynamic variables are still the most relevant predictors for long-term outcomes (as mirrored in the ELN recommendations) [2,9,10,11,12]. Recently, intracellular Separase proteolytic activity was identified as a potential marker of molecular response, *BCR::ABL1* positivity and proliferation of CD34+ cells in TKI-treated CML patients [13]. 

### 1.2. ESPL1/Separase, PTTG1/Securin and PTTG1IP/Securin Interacting Protein 

*ESPL1*/Separase, a cysteine endopeptidase, is a key player in chromosomal segregation. In mitotic anaphase, it accomplishes proteolytic cleavage of Cohesin, a “glue” multi-protein complex that holds sister chromatids together [14,15]. Furthermore, Separase maintains chromatin architecture, thereby mediating regulatory domain interactions within transcriptionally associated domains (TADs) [16]. This activity on interphase chromosomes involves controlling replication fork speed and DNA repair mechanisms, thereby preventing oncogenic transformation [17,18,19]. In human cancers, *ESPL1*/Separase is frequently overexpressed and associated with the emergence of aneuploidy, a hallmark of advanced human malignancies [20,21,22,23]. Consequently, Separase is considered an aneuploidy promoter that, when overexpressed, can function as an oncogene [24].

Proper temporal and spatial activation of Separase proteolytic activity is ensured by multiple inhibitory mechanisms; first of all, *PTTG1*/Securin binding [14,25,26,27]. Securin binds Separase in a chaperone-like manner and inhibits its proteolytic activity by both blocking substrate interaction and compelling an inactive conformational state [28,29]. For Separase activation, Securin is targeted for proteasomal degradation by the ubiquitin ligase APC (anaphase-promoting complex), resulting in the broad destruction of intracellular Securin. This mediates the degradation of Cohesin followed by autocatalytic inactivation and complete proteolysis of Separase itself [14,30]. Therefore, post-mitotic cells lack both functional Separase and Securin that must be expressed de novo before the next mitotic round can take place [26]. Securin and Securin-interacting protein (*PTTG1IP*) are important for the nuclear accumulation of newly synthesized Separase molecules [28]. *PTTG1IP* represents a multifunctional glycosylated type I integral membrane protein, which mediates the translocation of Securin into the nucleus [29,31]. Since the overexpression of *PTTG1IP* has been associated with dysregulated p53 functions, *PTTG1IP* has been identified as a putative oncogene, thereby consorting with Separase and Securin [32]. 

Here, we investigated the predictive value of *ESPL1*, *PTTG1* and *PTTG1IP* gene expression for achievement of MMR under TKI first-line therapy within 12 months in newly diagnosed CML patients (*n* = 97). We found that distinct expression levels of the three functionally associated genes can predict MMR non-achievement. Pathogenetic context and the potential benefit in risk stratification of CML patients eligible for TKI first-line treatment are discussed. 

## 2. Materials and Methods

### 2.1. Patients and Controls 

Our investigation represents a small experimental pilot study with highly limited retrospective cDNA samples collected between the years 2002 and 2017 from *BCR::ABL1* routine diagnostics carried out at the laboratory for leukemic diseases of the Department of Hematology and Oncology, Medical Faculty Mannheim, University of Heidelberg. All cDNA samples were remains of a regular *BCR::ABL1* monitoring procedure accompanying the TKI treatment of patients at our site. By default, excess samples were kept frozen at −20 °C until they were selected for this investigation. Since the majority of patients were included in clinical CML studies, comprehensive clinical and molecular data for each patient, stored in a local database (LeukDB), was available for this investigation. In total, 97 CML patients in chronic phase (CP) at the time of initial diagnosis (ID) and 51 healthy controls were investigated (Appendix A). Randomly selected samples were solely classified based on time until the achievement of MMR (=0.1%IS). No additional criteria were applied that may cause potential biases. Of these, 46 patients (mean age 58 y, range 14–85 y, 59% male) were classified as responders (=R, achievement of MMR within 12 months of TKI treatment). Fifty-one patients (mean age 58 y, range 26–87 y, 59% male) were classified as non-responders (=NR, no MMR within 12 months of TKI treatment). The mean time until the achievement of MMR for the R cohort was 5 months (±3 M). The mean time until MMR for the NR cohort was 26 months (±15 M). With 27 (53%), more than half achieved MMR after the first 12 months of TKI treatment, while almost half of the NRs (*n* = 24, 47%) never achieved MMR (refer to Appendix A).

The distribution of the *BCR::ABL1* fusion type variants e13a2 and e14a2 was 48%/30% for R (*n* = 46) and 41%/37% for the NR group (*n* = 50), respectively. Two patients of the NR group (4%) expressed e1a2 fusion type. The mean white blood cell counts at ID were 42,089 (range: 3400 to 316,000) for the R and 90,978 (range: 2000 to 404,000) for the NR cohort. A total of 91 of all CML patients (*n* = 97) were untreated at the time of sampling. The remaining six patient samples stem from a point before therapy restarted after a pause of at least 3 months once the patients relapsed from initial remission. A total of 63% of all patients received only one TKI during the first 12 months after diagnosis, 37% of all patients have undergone a combination or change of therapy. The majority of patients received imatinib as the only drug (32%), followed by nilotinib (23%), dasatinib (7%) and bosutinib (1%) (Appendix A). *BCR::ABL1* tumor load reported as *BCR::ABL1%*IS, where available at the time of ID, averaged out at 29% (range: 0.02% to 85%) for the R and 35% (range: 0.15% to 178%) for the NR cohort. None of the analyzed patients showed additional cytogenetic aberrations. The study has been approved by the institutional ethics committee (Medizinische Ethikkommission II, Medizinische Fakultät Mannheim, Ruprecht Karls-Universität Heidelberg, #2016-506N-MA). All patients provided written informed consent in accordance with the Declaration of Helsinki.

### 2.2. Sample Preparation and Quantification of BCR::ABL1 

Whole peripheral blood was subjected to red blood cell lysis, and the resulting white blood cell (WBC) pellets were used for total RNA extraction, as described recently [33,34,35]. WBC prepared from whole blood samples of 51 anonymous healthy donors served as experimental controls and were treated in the same way as the patient samples. Total RNA preparation from clinical samples (Maxwell^®^MDx technology, Promega, Mannheim, Germany), absolute quantification of *BCR::ABL1* transcript levels (TaqMan 7500 Fast Real-Time PCR System, ThermoFisherScientific/Applied Biosystems, Waltham, MA, USA) and data evaluation were performed as described previously [35]. *GUSB* served as the housekeeping gene in all qRT-PCR experiments.

### 2.3. Relative Quantification of ESPL1, PTTG1, PTTG1IP Transcript Levels

The LightCycler 480 platform (Roche Applied Science, Mannheim, Germany) was used for qRT-PCR of the *GUSB, ESPL1, PTTG1* and *PTTG1IP* transcript levels employing SYBR Green I Master Mix” (Roche Applied Science), and target-specific primer sets (Hs_ESPL1_1_SG QuantiTect, Cat. No. QT00027216; Hs_PTTG1IP_1_SG QuantiTect, Cat. No. QT00076601; Hs_PTTG1_1_SG QuantiTect, Cat. No. QT00044037) according to the manufacturer’s manual (Qiagen, Hilden, Germany). Experiments were performed in triplicate. Relative quantification (2^(−ΔΔCt)^ method) followed the method of Livak and Schmittgen from 2001 [36].

### 2.4. Statistics 

Our investigation represents a small experimental pilot study with preselected (R or NR) and highly limited retrospective cDNA samples collected for years from *BCR::ABL1* routine diagnostics at our site. All available samples that matched the R and NR criteria as outlined in the Materials and Methods section are included in the investigation. Since no sample/patient randomization was carried out, no sample size calculation analysis has been performed. Statistical calculations and a 3D scatter plot were performed using SAS version 9.4 (SAS Institute GmbH, Heidelberg) and SPSS version 27 (IBM, Ehningen, Germany), respectively. Unpaired (Student’s *t*-test) and paired analyses (Wilcoxon signed rank test) were employed to calculate the variation ranges between the relative qRT-PCR data. Logistic regression (Fisher’s scoring) and analysis of maximum likelihood estimates were used to test for the correlation of distance with response. Clinical data were compared using parametric tests (Student’s *t*-test) and non-parametric (Mann–Whitney U-test, Fisher test, and chi-squared test). For association with time non-parametric data, Spearman, Kruskal–Wallis and Mann–Whitney U-tests were used. Values of *p* < 0.05 were considered significant. 

## 3. Results

### 3.1. Relative Gene Expression of ESPL1, PTTG1 and PTTG1IP

We have investigated relative transcript levels of *ESPL1*, *PTTG1* and *PTTG1IP* in the WBC of 97 CML patients (R, *n =* 46; NR, *n =* 51) at the time of ID and of 51 healthy controls by means of qRT-PCR. The measured Ct values were normalized to the housekeeping gene *GUSB*. The resulting ΔCt data were subjected to statistical analysis. It is noted that the ΔCt values are inversely correlated to the respective real transcript levels. 

Analysis of variance (Table 1) revealed global *p*-values of *p* < 0.0001 (*ESPL1*), *p* = 0.0036 (*PTTG1*) and *p* = 0.1736 (*PTTG1IP*). Pairwise testing for *ESPL1* and *PTTG1* showed significant differences between CML groups (R, NR) and controls, i.e., elevated transcript levels (FC = fold change) in the CML groups when compared to the control group. *ESPL1* and *PTTG1* showed significantly higher respective transcript levels in the CML NR (mean ΔCt*^ESPL1^* = 4.53 ± 1.86, FC*^ESPL1^* = 4.40, ΔCt*^PTTG1^* = 3.41 ± 1.72, FC*^PTTG1^* = 3.64) and R (mean ΔCt*^ESPL1^* = 4.94 ± 01.51, FC*^ESPL1^* = 3.13, ΔCt*^PTTG1^* = 3.42 ± 0.78, FC*^PTTG1^* = 1.88) group when compared to the control group (mean ΔCt*^ESPL1^* = 5.89 ± 0.76, ΔCt*^PTTG1^* = 4.10 ± 0.65). There were no significantly different respective transcript levels of *PTTG1IP* in the CML NRs (mean ΔCt*^PTTG1IP^* = 2.87 ± 2.57; FC*^PTTG1IP^* = 3.05) or R (mean ΔCt*^PTTG1IP^* = 2.38 ± 1.69, FC*^PTTG1IP^* = 2.67) group than in the control group (mean ΔCt*^PTTG1IP^* = 3.07 ± 0.93). No significance was observed when the CML R and NR groups were compared. For detailed statistics, see Appendix A. Correlations between gene pairs (Pearson) showed significance between *ESPL1*/*PTTG1* for the NR (*r* = 0.58, *p* < 0.0001) and R (*r* = 0.55, *p* < 0.0001) cohorts and *PTTG1*/*PTTG1IP* (*r* = 0.52, *p* = 0.0001) for the NR cohort. 

### 3.2. Distance Analysis for Risk Stratification

In the search for a better discriminatory power (R vs. NR), we ordered sample data within the three-dimensional room by constructing centroid spheres, one for each cohort, as illustrated in Figure 1. The center of each centroid corresponds to the respective mean ΔCt values of *ESPL1* (x-axis), *PTTG1* (y-axis) and *PTTG1IP* (z-axis). Distance (=cut-off) analysis of each CML sample (represented by *xyz*-coordinates) with respect to the commonly calculated CML centroid center resulted in a significant difference in spatial distance for NR when compared to the R group (NR, 2.98, 95% confidence limits: 2.40 to 3.55 vs. R, 2.23, 95% confidence limits: 1.97 to 2.49, *t*-test (Satterthwaite approximation) *p* = 0.0187). Logistic regression and analysis of maximum likelihood estimates revealed a positive correlation of distance with MMR non-achievement (*p* = 0.0388, odds ratio 1.479, 95% Wald confidence limits: 1.020 to 2.143). Thus, the larger the calculated distance, the higher the probability for prognosis of “NRs”. Calculated cut-offs (distances), including sensitivities and specificities for all CML samples (*n =* 97), are given in Table 2. Accordingly, 5 NR (10%) with distance values >5.9 display a 75% probability for MMR non-achievement. For distance values >5.9810, the specificity is 100%, concurring with a probability for MMR non-achievement of 81%. Patients with a distance of 8 display ≥90% probability for MMR non-achievement, which applies to 6% (*n =* 3) of the observed NR. 

Since samples 93, 94, 95, 96 and 97 (as shown at the end of Table 2) that correspond to patients 34, 51, 22, 10 and 27 (shown in Appendix A), respectively, show cut-offs of ≥5.9180 and 100% specificity, we state that these five NR samples are “predictable”. No additional criteria were applied that may cause potential biases. For detailed statistics, see Appendix A. ROC curve analysis revealed the best possible overall diagnostic accuracy of combined *ESPL1*, *PTTG1* and *PTTG1IP* transcript level testing at cut-off 3, i.e.*,* 85% specificity concurring with 37% sensitivity (Appendix A).

### 3.3. Leukocyte Count at ID Differs between R and NR Cohort

Further analyses of variance with respect to basic and clinical parameters between the R and NR cohorts revealed no significant differences regarding age, sex or *BCR::ABL1* fusion type (Table 3). As to be expected, the time until the achievement of MMR showed significance (*p* < 0.001). Furthermore, approximately twice the number of leukocytes at ID were found in the NR cohort (mean 90,977.45 cells/µL) when compared to the R cohort (mean 42,089.13 cells/µL; *p* = 0.018). The *BCR::ABL1* quotients showed a similar strong trend (*p* = 0.067). The proportion of imatinib to administered TKIs in total (*n =* 96, see Appendix A) was higher (*p* = 0.007) in the NR cohort (68%) than in the R cohort (31%). 

### 3.4. ESPL1, PTTG1 and PTTG1IP Gene Expression Levels in R Cohort Correlate with Time until Achievement of MMR

We have investigated potential correlations between time until the achievement of MMR and the gene expression levels of *ESPL1*, *PTTG1* and *PTTG1IP* (Figure 2). For the R cohort, negative correlations were found for the ΔCt values of *ESPL1* (*p* = 0.0002, panel B) and *PTTG1* (*p* = 0.0114, panel D). Due to the inverse correlation, this indicates that a lower gene expression of the *ESPL1* and *PTTG1* correlates with a faster achievement of MMR. In contrast, ΔCt values of *PTTG1IP* (*p* = 0.0152, panel E) correlated positively with time until the achievement of MMR, indicating that a higher expression of *PTTG1IP* correlates with a faster response to therapy. In contrast, the NR cohort did not show any relation regarding the gene expression levels. 

### 3.5. Leukocyte Counts at Time of ID, BCR::ABL1 Quotients and TKI Therapy Correlate with Time until Achievement of MMR

We observed a significant correlation between the leukocyte count at the time of ID and the time until the achievement of MMR for the NR (Figure 3A) R cohort (*p* = 0.0038, Figure 3B). A significant linear correlation in the R cohort was found between *BCR::ABL1*% and the time until the achievement of MMR (*p* = 0.0025, Figure 3D) but not in the NR cohort (Figure 3C). This indicates that higher leukocyte counts and higher *BCR::ABL1* quotients at ID correlate with a slower response to TKI therapy. We further analyzed the 53 (55%) patients receiving monotherapy with nilotinib or imatinib during the first 12 months. We found a significant negative correlation between the time until the achievement of MMR and the choice of TKI therapy in the R cohort (*r* = −0.46, *p* = 0.019). No significant relationship between basic parameters (age, gender) and type of *BCR::ABL1* gene fusion with the time until the achievement of MMR could be detected. 

### 3.6. Predictable NR Display Lower ESPL1 and PTTG1IP Transcript Levels Compared to Corresponding R

Data comparison between the five predictable NR (patients no. 10, 22, 27, 34, 51, selected according to data shown in Figure 1 and Table 2) and the five fastest R patients (no. 1, 2, 3, 4, 5; compare Appendix A) revealed a significantly larger distance (*p* = 0.0028), lower *ESPL1* (*p* = 0.018) and *PTTG1IP* (*p* = 0.0168) transcript levels (higher ΔCt values) and a higher *BCR::ABL1* quotient (*p* = 0.0414) in the NR compared to the fastest R patients (Table 4). Two of the predictable NR never achieved MMR, and one predictable NR achieved MMR within the respective observation period (median 23, range 16 to 91 months), whereas all of the five fastest R achieved MR4 at minimum. 

### 3.7. NR and R Cohort Assignment Concurs with Rate of BCR::ABL1 Decline after 3 Months of TKI Treatment

Recently, the rate of *BCR::ABL1* decline at 3 months has been reported as a critical prognostic discriminator of CML patients [10]. In order to check the validity of our NR and R cohorts, we have calculated the rate of *BCR::ABL1* decline from baseline (time of ID) with respect to *BCR::ABL1%*IS load at 3 months resulting in the *BCR::ABL1* halving time (in days). As depicted in Table 5, we found that the halving time of the NR cohort (mean 86, range 45–541) differed significantly (*p =* 0.0176) from that of the R cohort (mean 51, range 45–133). It is to note that for 9 patients of the NR cohort and for 4 patients of the R cohort, increased values were found after 3 months of TKI treatment (maybe due to resistance or bad compliance). For these, doubling times were calculated and listed separately. Here, a trend of increased doubling times for the R cohort (mean 268) was found when compared to the NR cohort (mean 72), pointing to lower proliferation rates of tumor cells within the R cohort. 

Our findings concur with the observations of Branford and others that the rate of *BCR::ABL1* decline can serve as a critical prognostic discriminator of CML patients after the first 3 months of TKI treatment [10].

## 4. Discussion

In this study, we demonstrate that the gene expression levels of *ESPL1*/Separase and two functionally associated genes, *PTTG1*/Securin and *PTTG1IP* (Securin interacting protein), can have a predictive value for achievement of MMR under TKI first-line therapy within 12 months for a distinct set of newly diagnosed CML patients. Several lines of evidence point to the beneficial value of gene expression level testing at the time of diagnosis. 

First, we found that *ESPL1* and *PTTG1* are expressed at higher transcript levels in CML patients than in the control group. Furthermore, the observed correlations between the expression levels of *ESPL1*/*PTTG1* and *PTTG1/PTTG1IP* confirm the state of knowledge about the functional interaction of these proteins on a genetic level also for CML patients [28,37]. This is the first comprehensive examination of transcript levels of *ESPL1, PTTG1* and *PTTG1IP* in CML patients, and the observed overexpression confirms the role of *ESPL1* as a surrogate marker of proliferation in CML [38,39]. Our findings further support the hypothesis that increased activity of Separase in CML patients may concur with an altered expression of *ESPL1* [21]. Our gene expression data are consistent with diagnostic clinical parameters as higher leukocyte counts and higher *BCR::ABL1* quotients at ID are found in the NR cohort and concur with a slow response to therapy. The most plausible explanation for these correlating observations is that CML patients at the time of ID are characterized by more or less severe leucocytosis and display a high proportion of *BCR::ABL1* positive peripheral blood cells with high proliferation potential and dysfunctional cell differentiation. Therefore, the higher the tumor load, as indicated by *BCR::ABL1%*, the longer an administered TKI therapy will take to reduce tumor cell burden to MMR levels (=0.1% *BCR::ABL1* IS). 

Second, we found a significant correlation between the time until the achievement of MMR and the relative transcript levels of *ESPL1*, *PTTG1* and *PTTG1IP* in the R cohort. Our results indicate that lower expressions of *ESPL1* and *PTTG1* at ID (as indicated by high ΔCt values) are associated with a shorter time until the achievement of MMR in the R cohort and hence a better response to TKI therapy. To our knowledge, this is the first report of this correlation in CML patients. These findings are in line with previous observations that linked an overexpression of *ESPL1*/Separase and *PTTG1*/Securin to a less favorable tumor classification [24,40]. Valid correlations could not be stated for the NR cohort, most likely due to vastly scattering data (time until the achievement of MMR) within the small cohort (*n* = 51). 

Our results that the observed lower expression levels of *ESPL1* and *PTTG1* at ID concur with faster response to TKI therapy are not valid for the five predictable NR where an inverse correlation has been found (Table 4). Here, lower *ESPL1* and *PTTG1IP* transcript levels (as indicated by higher ΔCt values) were displayed by the predictable NR (*n* = 5) when compared to the five fastest R. However, leukocyte counts (trend) and *BCR::ABL1*% quotient are in line with the respective R and NR overall data set. It is currently unclear what molecular mechanisms or genetic alterations (may be due to genetic predisposition) may be responsible for the unexpected gene expression pattern of the five NR, which can also obviously serve as a marker for MMR prediction. Potential explanations are offered in previous studies indicating a correlation between the downregulation of Separase, polyploidy and the development of different cancer entities [26,41]. Furthermore, one could imagine that a compensatory mechanism that elevates the proteolytic activity of Separase in NR after TKI induced decrease of *ESPL1* expression may lead to a delayed response to TKI therapy and a worse course of the disease as assumed recently [39]. On the other hand, depletion of Separase may prevent the segregation of sister chromatids leading to Structural maintenance of chromosomes protein 3 (SMC3) acetylation and a failure in blocking the cell cycle during replication [18]. Resulting in erratic replication rounds would be in line with the higher leukocyte counts and elevated *BCR::ABL1* quotients in the predictable NR patients. 

For the overall R cohort (Figure 2F), our data correlate high *PTTG1IP* expression levels (as indicated by low ΔCt values) with fast response to TKI therapy. This is in accordance with the *PTTG1IP* expression data of the five predictable NR (Table 4) that showed significantly smaller transcript levels than the fastest R. These findings are in line with previous observations linking underexpression of *PTTG1IP* and a higher expression of Securin with a higher risk in breast cancer patients for a more aggressive progression. Mechanistically, an underexpression of *PTTG1IP* may result in a missing nuclearization of Securin, leading to aberrant high Securin levels located in the cytoplasma and a lack of proper Separase control [42]. The resulting uncontrolled proteolytic activity of Separase and unscheduled replication rounds would be in line with higher leukocyte counts and elevated *BCR::ABL1* quotients observed in the NR patient cohort.

It is unquestionable that TKI therapy regimen, i.e., type of initial TKI, cytoreductive (HU, AraC) or immune-modulating (IFN-alpha) pre-therapies play a considerable role in the time interval until the achievement of MMR. In our study, fewer patients of the R cohort received imatinib as first-line TKI (compare Appendix A). Additionally, we found a negative correlation in the R cohort between time until the achievement of MMR and choice of TKI with regards to those patients (55% of patients) receiving only imatinib or nilotinib (monotherapies) within the first 12 months after diagnosis. This may indicate that 2nd generation TKIs lead to faster achievement of MMR, congruent with past observations showing that patients receiving first-line nilotinib show better response with higher rates of cytogenetic and molecular remissions [2]. However, due to the high heterogeneity of TKI therapies within the relatively small R and NR cohorts, we cannot make valid statements about the influence of the various TKIs on the predictability of MMR achievement within 12 months of treatment.

One could argue that the accuracy of conventional *BCR::ABL1* minimal residual disease assessment may depend on the applied PCR technique as RT-qPCR has been reported as amplifying b3a2 and b2a2 with different performance resulting in underestimation of the e14a2 variant compared to the e13a2 variant [43,44]. Digital PCR (dPCR) may overcome this limit as it is independent of amplicon length. In this study, the diagnostic status “MMR” was assessed according to the international guidelines of Foroni et al. 2011 [45]. The robustness and comparability of the diagnostic data measured here have been repeatedly confirmed by international harmonization trials within the European Treatment and Outcome Study for CML (EUTOS) [46]. Between 2016 and 2021, we have been using cell-based *BCR::ABL1* reference panels traceable to the World Health Organization primary reference material to standardize and validate our local laboratory qRT-PCR test system and the corresponding validated conversion factors (CFs) to the International Scale as described previously [35]. Therefore, we consider the influence of the applied method on the observed findings to be negligible. 

However, it is to note that the findings of our investigation strongly depend on the qRT-PCR methodology used here. It is conceivable that methods with the potential for higher sensitivity, such as digital PCR (dPCR), may shift the diagnostic log level classification for a minor subset of NR patients. Therefore, the assignment of our patient collective to NR or R is definitely valid only in combination with the RT-qPCR method applied in this investigation.

Third, the prognostic value is the spatial distance of a patient’s data set to a defined centroid (depicted in Figure 1) when ΔCt values of the three genes are arranged in an *xyz*-coordinate system. Data evaluation can easily be automated and implemented into routine laboratory *BCR::ABL1* monitoring algorithms. Since resistance to the initial TKI therapy occurs in 10–15% of patients receiving imatinib and in about 10% of patients receiving therapy with second-generation TKIs as first-line treatment [2], patients could benefit from an improved risk stratification instead of losing valuable time with therapy changes and preventable progression of the disease. Furthermore, the economic burden could be considerably lowered (e.g., in Germany, €48,000–81,000 annually) [47], emotional suffering prevented and compliance with therapy improved.

Recently, TKI response-related dynamic variables, such as the rate of *BCR::ABL1*, decline at 3 months of TKI treatment was reported as a critical prognostic discriminator of CML patients [2,9,10,11,12]. The calculation of *BCR::ABL1* decline rates (halving times) for our NR and R cohorts confirm the observations of Branford and others that the rate of *BCR::ABL1* decline can serve as a prognostic discriminator of CML patients after the first 3 months of TKI treatment [10]. However, patients have to undergo 3 months of TKI treatment before prognostic statements can be made. Therefore, a qRT-PCR-based assay that can be applied to patients at the time of ID may be a valuable addition to the existing set of diagnostic tools. This may contribute to a more personalized TKI therapy in CML, albeit the benefit may be limited to 10% of patients.

Despite the poor performance in terms of the separation of overall R and NR cohorts, 10% of the observed NR could have been reliably predicted at the time of diagnosis. At a distance value of 5.9 and with a probability of 75%, an insufficient response to TKI therapy is very likely to advocate for the initiation of more individualized therapy. With increasing distance, it becomes more and more likely to predict a NR since the probability increases by 48% when the distance is increased by one. Accordingly, patients with a distance of 8 display ≥90% probability of MMR failure, which applies to 6% of the observed NR. Thus, a distinct subset of CML patients lacking response to an inadequate TKI therapy could be identified already at the time of diagnosis. On the other hand, with the suggested cut-off at 3, about 37% of therapy failures could already be identified at the time of diagnosis, while less than one-fifth of R would mistakenly be classified as a treatment failure. One could argue that only a rather small portion of all CML patients would benefit from this prediction tool, as for patients with a distance value <5, a safe classification into a response group is not possible. However, the novel distance score could still be used as a tendency indicator of TKI response and a useful tool on the way toward individualized therapy in CML. A prospective controlled and randomized study with larger cohorts should be performed to measure the reliability of the proposed cut-offs and to check the applicability of the qRT-PCR assay in CML routine diagnostics. 

## 5. Conclusions

In conclusion, we have demonstrated that the analysis of *ESPL1*, *PTTG1* and *PTTG1IP* gene expression levels by qRT-PCR in the peripheral blood of patients at the time of ID can contribute to an early individualized TKI therapy in CML. Our established novel distance (cut-off) score based on *ESPL1*, *PTTG1* and *PTTG1IP* gene expression levels can serve as a predictor of TKI non-response and a tool in risk stratification of CML patients before initiation of TKI first-line treatment.

## Figures and Tables

**Figure 1 cancers-15-02652-f001:**
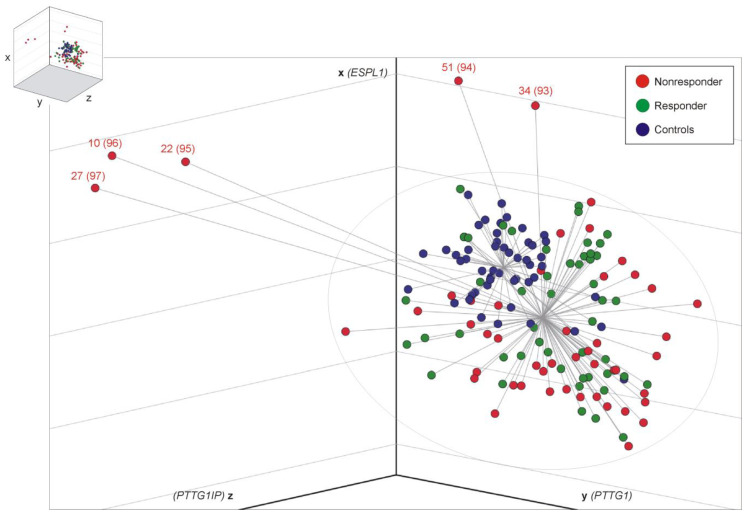
3D scatter plot illustration of 148 analyzed samples. Relative transcript level data (ΔCt values) of the genes *ESPL1* (*x*-axis), *PTTG1* (*y*-axis) and *PTTG1IP* (*z*-axis) was arranged in the three-dimensional room with a center for healthy controls (blue, *n* = 51) and a common center for CML R (green, *n* = 46) and NR samples (red, *n* = 51). Logistic regression and analysis of maximum likelihood estimates revealed a positive correlation of distance with MMR failure (*p =* 0.0388, odds ratio 1.479, 95% Wald confidence limits: 1.020 to 2.143). The ellipsoid in grey symbolizes the cut-off (distance) of 5.9180; then, the test reached 100% specificity, as outlined in Table 2. Therefore, about 10% (5 of 51) of the NR samples could have been predicted correctly at the time of ID. The first numbers refer to the patient data in Appendix A, and the numbers in brackets refer to Table 2. Statistical analysis: the MEANS, TTEST (Satterthwaite approximation), and LOGISTIC (Fisher’s scoring, Wald testing) procedures used SAS Version 9.4.

**Figure 2 cancers-15-02652-f002:**
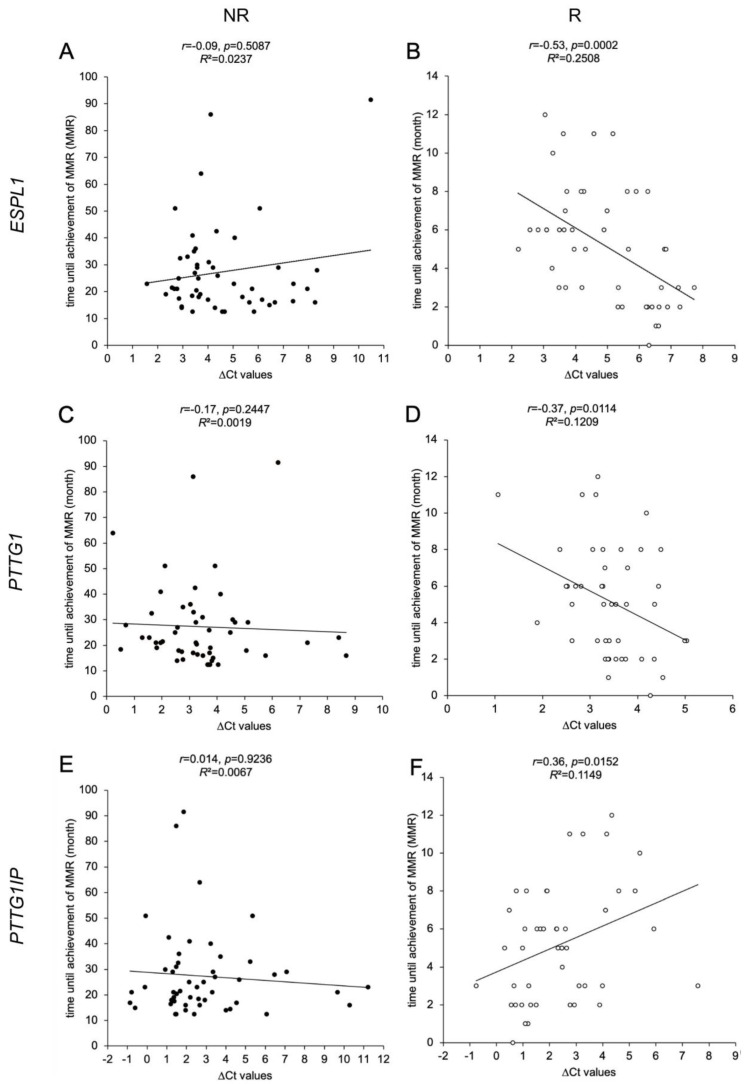
Correlations of *ESPL1*, *PTTG1* and *PTTG1IP* expression levels with time until the achievement of MMR in CML patients. Scatter plot illustrations of 97 analyzed samples (non-responders (NR) as black dots, *n* = 51, (**A**,**C**,**E**); responders (R) as white dots, *n* = 46, (**B**,**D**,**F**)). The illustrations show the association between time until the achievement of MMR (*y*-axis in A-F) in CML patients with the calculated ΔCt values that serve as a measure of relative gene expressions of *ESPL1*/Separase (x-axis; (**A**,**B**)), *PTTG1*/Securin (*x*-axis; (**C**,**D**)), *PTTG1IP* (*x*-axis; (**E**,**F**)). It is noted that the ΔCt values are inversely correlated to the respective real transcript levels. Each data point in the illustration represents the mean ΔCt value of triplicate measurements. Time until the achievement of MMR was derived from the patient database (LeukDB) of the laboratory for leukemia diagnostics of the III. Medical University Clinic Mannheim. Statistical analysis: MEANS, Spearman correlation coefficient, effect size *r*, and coefficient of determination *R*^2^.

**Figure 3 cancers-15-02652-f003:**
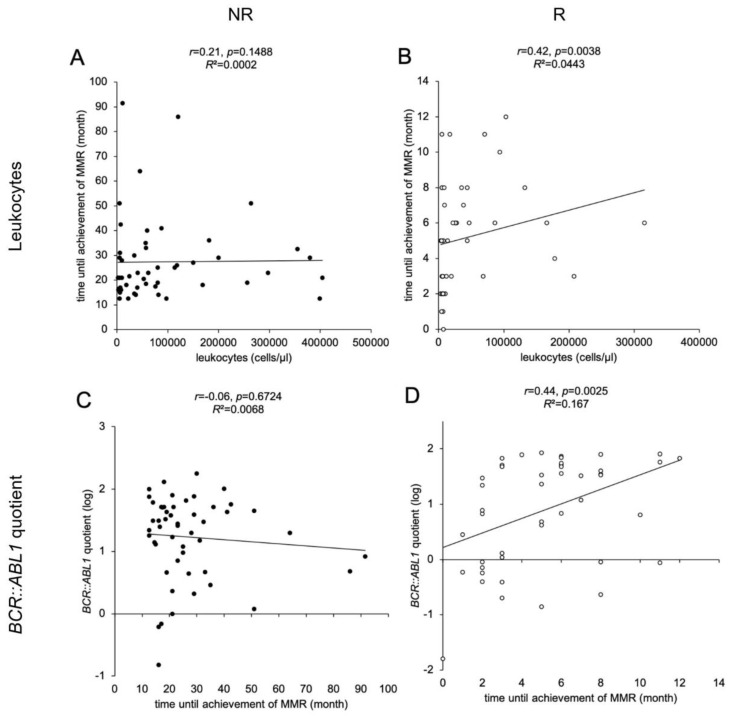
Correlations of leukocyte counts and *BCR::ABL1* quotients with time until the achievement of MMR in CML patients. Scatter plot illustrations of 97 analyzed samples (non-responders (NR) as black dots, *n* = 51, (**A**,**C**)); responders (R) as white dots, *n* = 46, (**B**,**D**)). The illustrations show the association between time until the achievement of MMR (*y*-axis in A, B; *x*-axis in (**C**,**D**)) in CML patients with leukocytes at ID (*x*-axis; (**A**,**B**)) and *BCR::ABL1* quotient at ID (*y*-axis; (**C**,**D**)). Each data point in the illustration represents one examined patient. Time until the achievement of MMR, leukocyte count at ID and *BCR::ABL1* quotient were derived from the patient database (LeukDB) of the laboratory for leukemia diagnostics of the III. Medical University Clinic Mannheim. Statistical analysis: MEANS, Spearman correlation coefficient, effect size *r*, and coefficient of determination *R*^2^.

**Table 1 cancers-15-02652-t001:** ANOVA and pairwise testing for relative transcript levels (ΔCt values) of *ESPL1*, *PTTG1* and *PTTG1IP* within the R, NR and control groups.

ANOVA	*ESPL1*	*PTTG1*	*PTTG1IP*
global	*p* < 0.0001	*p =* 0.0036	*p =* 0.1736
controls vs. NR *	*p* < 0.0001	*p =* 0.0127	-
controls vs. R *	*p =* 0.0061	*p =* 0.0167	-
NR vs. R *	*p =* 0.3910	*p =* 0.9998	-

* Post-hoc pairwise testing according to Scheffé.

**Table 2 cancers-15-02652-t002:** Cut-offs (distances), sensitivities and specificities for CML samples tested (*n* = 97).

Sample No.	Sensitivity	Specificity	Sum	Cut-Off
1	1	0	1	0.549594
2	1	0.021739	1.021739	0.633119
3	1	0.043478	1.043478	0.750284
4	1	0.065217	1.065217	0.773095
5	1	0.086957	1.086957	0.951466
6	1	0.108696	1.108696	0.980357
7	0.980392	0.108696	1.089088	1.150964
8	0.960784	0.108696	1.06948	1.156968
9	0.960784	0.130435	1.091219	1.162309
10	0.960784	0.152174	1.112958	1.17719
11	0.941176	0.152174	1.09335	1.244523
12	0.921569	0.152174	1.073743	1.335997
13	0.901961	0.152174	1.054135	1.355251
14	0.882353	0.152174	1.034527	1.441234
15	0.882353	0.173913	1.056266	1.488975
16	0.862745	0.173913	1.036658	1.534161
17	0.843137	0.173913	1.01705	1.601827
18	0.823529	0.173913	0.997442	1.605338
19	0.823529	0.195652	1.019182	1.61878
20	0.823529	0.217391	1.040921	1.637496
21	0.803922	0.217391	1.021313	1.647798
22	0.803922	0.23913	1.043052	1.652854
23	0.784314	0.23913	1.023444	1.708054
24	0.764706	0.23913	1.003836	1.757046
25	0.764706	0.26087	1.025575	1.793212
26	0.745098	0.26087	1.005968	1.793669
27	0.72549	0.26087	0.98636	1.79974
28	0.705882	0.26087	0.966752	1.810044
29	0.705882	0.282609	0.988491	1.840533
30	0.686275	0.282609	0.968883	1.864345
31	0.686275	0.304348	0.990622	1.95028
32	0.666667	0.304348	0.971014	1.964042
33	0.666667	0.326087	0.992754	2.015026
34	0.666667	0.347826	1.014493	2.039745
35	0.647059	0.347826	0.994885	2.042153
36	0.627451	0.347826	0.975277	2.05187
37	0.627451	0.369565	0.997016	2.057197
38	0.607843	0.369565	0.977408	2.069368
39	0.607843	0.391304	0.999147	2.085874
40	0.588235	0.391304	0.97954	2.096926
41	0.568627	0.391304	0.959932	2.122027
42	0.568627	0.413043	0.981671	2.124711
43	0.568627	0.434783	1.00341	2.180675
44	0.568627	0.456522	1.025149	2.189591
45	0.54902	0.456522	1.005541	2.195555
46	0.54902	0.478261	1.02728	2.20937
47	0.54902	0.5	1.04902	2.277969
48	0.54902	0.521739	1.070759	2.283053
49	0.54902	0.543478	1.092498	2.316287
50	0.54902	0.565217	1.114237	2.358355
51	0.54902	0.586957	1.135976	2.363681
52	0.529412	0.586957	1.116368	2.376338
53	0.509804	0.586957	1.09676	2.405917
54	0.509804	0.608696	1.1185	2.429324
55	0.509804	0.630435	1.140239	2.444764
56	0.509804	0.652174	1.161978	2.456408
57	0.509804	0.673913	1.183717	2.465865
58	0.490196	0.673913	1.164109	2.469809
59	0.470588	0.673913	1.144501	2.516639
60	0.45098	0.673913	1.124893	2.564055
61	0.45098	0.695652	1.146633	2.565945
62	0.45098	0.717391	1.168372	2.611456
63	0.431373	0.717391	1.148764	2.678705
64	0.431373	0.73913	1.170503	2.710391
65	0.431373	0.76087	1.192242	2.73141
66	0.411765	0.76087	1.172634	2.771444
67	0.392157	0.76087	1.153026	2.799222
68	0.392157	0.782609	1.174766	2.815116
69	0.392157	0.804348	1.196505	2.85643
70	0.392157	0.826087	1.218244	2.860967
71	0.372549	0.826087	1.198636	2.880366
72 *	0.372549	0.847826	1.220375	3.021803
73	0.352941	0.847826	1.200767	3.028484
74	0.333333	0.847826	1.181159	3.061431
75	0.313725	0.847826	1.161552	3.115181
76	0.313725	0.869565	1.183291	3.149723
77	0.313725	0.891304	1.20503	3.156747
78	0.294118	0.891304	1.185422	3.205288
79	0.27451	0.891304	1.165814	3.209935
80	0.27451	0.913043	1.187553	3.334363
81	0.254902	0.913043	1.167945	3.347152
82	0.235294	0.913043	1.148338	3.452303
83	0.235294	0.934783	1.170077	3.481048
84	0.215686	0.934783	1.150469	3.481381
85	0.215686	0.956522	1.172208	3.547112
86	0.215686	0.978261	1.193947	3.579134
87	0.196078	0.978261	1.174339	3.624489
88	0.176471	0.978261	1.154731	3.698418
89	0.156863	0.978261	1.135124	3.789116
90	0.137255	0.978261	1.115516	3.802186
91	0.117647	0.978261	1.095908	4.781442
92	0.098039	0.978261	1.0763	5.099709
93 **	0.098039	1	1.098039	5.917999
94	0.078431	1	1.078431	6.435286
95	0.058824	1	1.058824	8.635014
96	0.039216	1	1.039216	9.944185
97	0.019608	1	1.019608	10.28212

* Best cut-off is reached when the sum of specificity and sensitivity reaches a maximum. ** For samples with cut-offs ≥5.9180, the specificity is 100%.

**Table 3 cancers-15-02652-t003:** Analyses of variance of basic and clinical parameters in CML R vs. NR patients (*n* = 97).

Parameter	Test	*p*-Value
Age	*t*-test	0.873
Sex	Chi^2^-test	0.990
Time until achievement of MMR	U-test	<0.001
Leukocyte count at ID	U-test	0.018
*BCR::ABL1* gene fusion type	Fisher-test	0.487
*BCR::ABL1* quotient	U-test	0.067
*BCR::ABL1* quotient IS *	U-test	0.173
Proportion of imatinib to administered TKIs	U-test	0.007

* *BCR::ABL1%*IS values only available for *BCR::ABL1* < 10% (*n =* 37).

**Table 4 cancers-15-02652-t004:** Comparison of the predictable NR (*n* = 5) vs. the fastest R patients (*n* = 5).

	NR	R	*p*-Values
Distance *	8.42 ± 1.99	2.47 ± 0.18	0.0028
ΔCt ** *ESPL1*	8.48 ± 1.17	6.52 ± 0.25	0.018
ΔCt ** *PTTG1*	6.26 ± 3.26	3.97 ± 0.57	0.1936
ΔCt ** *PTTG1IP*	7.89 ± 3.82	1.32 ± 0.84	0.0168
Leukocytes (cells/μL)	19760 ± 23302	7180 ± 2097	0.2946
*BCR::ABL1* quotient (%)	14.29 ± 10.16	0.88 ± 1.10	0.0414

* To common centroid of CML patients (NR and R). ** High ΔCt values mean low real transcript levels.

**Table 5 cancers-15-02652-t005:** Rates of *BCR::ABL1* decline (halving time) at 3 months after TKI treatment for NR (*n* = 49) and R (*n* = 45) cohorts.

	NR (*n* = 49)		R (*n* = 45)	
	Halving Time (Days) * 3 M (*n* = 40)	Doubling Time (Days) * 3 M (*n* = 9)	Halving Time (Days) * 3 M (*n* = 41)	Doubling Time (Days) * 3 M (*n* = 4)
Mean	86 ± 87	72 ± 55	51 ± 18	268 ± 280
Median	55	52	45	223
Min	45	11	45	26
Max	541	186	133	600
Range	496	175	88	574

* See Appendix A for raw data and sample selection.

## Data Availability

All data generated in this study has been included in the published article and/or the respective Appendix A.

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
