# Peer review of "Gene Expression Pattern of ESPL1, PTTG1 and PTTG1IP Can Potentially Predict Response to TKI First-Line Treatment of Patients with Newly Diagnosed CML"

_cancers, 2023, doi:10.3390/cancers15092652_

Round 1
Reviewer 1 Report (Previous Reviewer 1)
The methodological issue I pointed out in the initial review has been resolved. Therefore, the results obtained in this study and the authors' discussion are considered valid.
Author Response
Reviewer I
Comments and Suggestions for Authors
The methodological issue I pointed out in the initial review has been resolved. Therefore, the results obtained in this study and the authors' discussion are considered valid.
Reply
We thank reviewer I for his/her helpful comments/suggestions and the fair reviewing process. Thus, the manuscript was significantly improved.
Reviewer 2 Report (Previous Reviewer 2)
The authors have made significant improvements by addressing most of the comments raised in the previous report. While many points have been corrected and improved, some of the suggested experiments could not be completed due to limitations in their research center. Based on the authors' work, I believe the article has the potential to be accepted with a few changes.
I recommend revising the title, which may be considered misleading. Although the article suggests that the expression patterns of ESPL1, PTTG1, and PTTG1IP genes may have the potential to predict the response to TKI, the evidence presented is not sufficient to make a definitive statement. Therefore, I suggest modifying the title to reflect the potential rather than certainty of the gene expression patterns in predicting TKI response.
I appreciate the authors' efforts to address the previous comments. However, I still have some concerns regarding the selection of the five predictable NR patients (no. 10, 22, 27, 34, 51) mentioned in line 343. While statistical criteria can be used to identify patients with certain characteristics, it is important to provide a clear rationale for their selection.
Therefore, I suggest the authors provide additional details on the selection process of these patients. Specifically, they should explain why these five patients were chosen, whether there were any additional criteria, and if there were any potential biases in their selection.
No further comments.
Author Response
Reviewer II
Comment 1
The authors have made significant improvements by addressing most of the comments raised in the previous report. While many points have been corrected and improved, some of the suggested experiments could not be completed due to limitations in their research center. Based on the authors' work, I believe the article has the potential to be accepted with a few changes.
I recommend revising the title, which may be considered misleading. Although the article suggests that the expression patterns of ESPL1, PTTG1, and PTTG1IP genes may have the potential to predict the response to TKI, the evidence presented is not sufficient to make a definitive statement. Therefore, I suggest modifying the title to reflect the potential rather than certainty of the gene expression patterns in predicting TKI response.
Reply 1
Following the request of reviewer II we have changed now the manuscript title from “Gene expression pattern of ESPL1, PTTG1 and PTTG1IP can predict response to TKI first line treatment of patients with newly diagnosed CML” into “Gene expression pattern of ESPL1, PTTG1 and PTTG1IP can potentially predict response to TKI first line treatment of patients with newly diagnosed CML”. A “potentially” was inserted on page 1 line 3 as suggested to weaken the statement of the manuscript.
Comment 2
I appreciate the authors' efforts to address the previous comments. However, I still have some concerns regarding the selection of the five predictable NR patients (no. 10, 22, 27, 34, 51) mentioned in line 343. While statistical criteria can be used to identify patients with certain characteristics, it is important to provide a clear rationale for their selection. Therefore, I suggest the authors provide additional details on the selection process of these patients. Specifically, they should explain why these five patients were chosen, whether there were any additional criteria, and if there were any potential biases in their selection.
Reply 2
The five predictable NR patients (no. 10, 22, 27, 34, 51) mentioned in line 340 (former line 343) were part of the NR cohort (n=51). All NR were randomly selected patient samples from our biobank that were solely classified based on time until achievement of MMR (=0.1%IS). No additional criteria have been applied that may cause potential biases. Thus, the five predictable NR patients (no. 10, 22, 27, 34, 51) were selected the same way as all other 46 NR patients. At the time of selection there was no clue/evidence of predictability at all. Therefore, we cannot provide any other detail on the selection process of these 5 patients as given in section “2.1. Patients and controls” (page 3 lines 115 to 134). To clear up any potential misunderstandings we have inserted a statement on criteria and potential bias in lines 125 to 127.
The selection of the 5 predictable NR shown in Table 4 (line 349) is the result of our distance analysis, the result of which is graphically shown in figure 1 and described on page 5 lines 230 to 235 (see also end of Table 2). No additional criteria have been applied that could cause potential biases. To emphasize this, we inserted the following statement on page 6 lines 236 to 238.
Since the samples 93, 94, 95, 96 and 97 (as shown at the end of Table 2) that correspond to patients no 34, 51, 22, 10 and 27 (shown in Supplementary file 1), respectively, show cut-offs ≥ 5.9180 and 100% specificity, we state these 5 NR samples “predictable”. No additional criteria have been applied that may cause potential biases.
We further better refer to Figure 1 and Table 2 on page 13 lines 340 to 341.
Reviewer 3 Report (Previous Reviewer 3)
The Manuscript by Christiani and Colleagues has been partially improved from the previous submission. I have still some doubts about the study design and the analysis. I suggested the Authors a simple analysis about the slope of the MRD by considering the MR expressed as %IS instead of MR classes, but the authors did not performed it. This analysis could confirm the results presented by the authors, despite the lack of accuracy of the RT-qPCR. The data are available by the Authors, since the %IS is mandatory. The analysis of the slope of the MRD during the TKI therapy is very interesting and could potentially give additional sparks or carry new insights.
The English Language is very good.
Author Response
Reviewer III
Comment 1
The Manuscript by Christiani and Colleagues has been partially improved from the previous submission. I have still some doubts about the study design and the analysis. I suggested the Authors a simple analysis about the slope of the MRD by considering the MR expressed as %IS instead of MR classes, but the authors did not perform it. This analysis could confirm the results presented by the authors, despite the lack of accuracy of the RT-qPCR. The data are available by the Authors, since the %IS is mandatory. The analysis of the slope of the MRD during the TKI therapy is very interesting and could potentially give additional sparks or carry new insights.
Reply 1
As requested by Reviewer 3 we have performed now analysis of the BCR::ABL1 slope considering the MR expressed as %IS according to the recent study of Branford et al., 2014 (cited as reference [10]).
We have calculated the rate of BCR::ABL1 decline from baseline (time of initial diagnosis (ID)) with respect to BCR::ABL1%IS load at 3 months resulting in the BCR::ABL1 halving time (in days). We found that the halving time of the NR cohort (mean 86, range 45-541) differed significantly (p=0.0176) from that of the R cohort (mean 51, range 45-133).
Therefore, we confirm the observation of Branford and others that the rate of BCR::ABL1 decline can serve as critical prognostic discriminator of CML patients after the first 3 months of TKI treatment.
However, it is to note that for 9 patients of the NR cohort and for 4 patients of the R cohort increased values were found after 3 months of TKI treatment (maybe due to resistance or bad compliance). For these, doubling times were calculated and listed separately. Here, a trend to increased doubling times for the R cohort (mean 268) was found when compared to the NR cohort (mean 72).
Inclusion of the new data led to the extension of the manuscript.
i) Table 5 and accompanying text was inserted on page 13 line 353 to page 14 line 382.
ii) The corresponding raw data was given in Supplementary file S5.
iii) A short paragraph was added to the section “4. Discussion” (page 16 lines 491 to 500) to address the impact of our findings and the new slope data, i.e. confirmation of the observation made by Branford and others.
Round 2
Reviewer 3 Report (Previous Reviewer 3)
The Authors addressed all my comments and improved the quality of the reported data.
The English language is fine. There are just some typo errors here and there.
This manuscript is a resubmission of an earlier submission. The following is a list of the peer review reports and author responses from that submission.
Round 1
Reviewer 1 Report
This study investigates the mRNA expression of ESPL1, PTTG1, and PTTG1IP in peripheral blood mononuclear cells of chronic phase CML at diagnosis. Expression of ESPL1 and PTTG1 was higher in CML patients than in healthy subjects, especially in patients who did not achieve an optimal response (MMR by 12M). It is concluded that these gene expressions are valuable predictors of TKI response.
Although the findings obtained from this study are of academic interest, there are some concerns regarding the correct interpretation of the results. Therefore, please refer to the comments below.
Comments)
1. In the chronic phase of CML, leukemic cells are mainly distributed in the granulocytic fraction, and the mononuclear cell fraction may or may not contain Ph-positive cells. (Primo et al. Br J Haematol 2006;132:736) It is unclear whether the gene expression in this study targets tumor cells or normal cells. Please clarify why mononuclear cells were used as a sample for this analysis.
2. Please ensure that "mononuclear cells" separated from blood samples of healthy individuals were used as control. If not, those gene expressions in the control samples need to be reevaluated via mononuclear cell separation.
3. Patients with IS ≤ 0.1% at diagnosis are included in this study. Please specify how this patient was dealt with for this analysis.
4. Please indicate the AUC value of the ROC curve (Figure S4). With a sensitivity of 37% and a specificity of 85%, these associations do not appear to be closely correlated.
Reviewer 2 Report
The study by Christiani et al. provides novel insights into the expression and prognostic value of ESPL1/Separase, PTTG1/Securin, and PTTG1IP/Securin interacting protein transcript levels in peripheral blood mononuclear cells of chronic myeloid leukemia (CML) patients at the time of initial diagnosis. The authors analyzed patient data from LeukDB and performed qRT-PCR assays to show that monitoring ESPL1, PTTG1, and PTTG1IP gene expression levels could aid in early individualized tyrosine kinase inhibitor (TKI) therapy for CML.
Although the manuscript presents a compelling case for the use of an ESPL1, PTTG1, and PTTG1IP gene expression score to predict TKI non-response, the authors suggest that further studies with larger cohorts and controlled, randomized designs are necessary to verify the reliability of the proposed cut-offs and the applicability of the qRT-PCR assay in routine CML diagnostics. The proposed ESPL1, PTTG1, and PTTG1IP gene expression score and qRT-PCR assay may have potential in predicting TKI non-response in CML patients, but it is essential to validate these findings in larger, diverse patient populations with standardized and controlled study designs. This will help establish the robustness of the proposed cut-offs and ensure that the assay can be reliably used in routine clinical settings.
After carefully reviewing the paper, I have identified several critical mistakes that require attention. For instance, the caption and legend for figure 2 are missing, and the description for figure 1 is mistakenly repeated. Additionally, there is a frequent error in the use of terminology where the scatter plots are referred to as scatter "blot." The materials and methods section lacks a detailed description of the criteria for patient and sample selection, particularly for the predictable NR analysis. This is a significant issue as the validity and generalizability of the findings depend on the criteria used. Furthermore, the absence of trend lines in the graphs makes it difficult to discern the strength and direction of correlations. The inclusion of trend lines is crucial to enhancing the clarity and interpretability of the findings. These concerns need to be addressed to improve the quality and validity of the study.
While the study is interesting, it is too speculative and descriptive. The manuscript would benefit from additional experiments that further explore the observed lower expression levels of ESPL1 and PTTG1 at initial diagnosis and their correlation with faster responses to TKI therapy.
As a result of these criticisms, I believe the manuscript is not yet suitable for publication.
Reviewer 3 Report
The manuscript presented by Christiani and colleagues reports a very interesting study among the expression of potential markers of response (meaning achievement of MMR within 12 months of treatment) in adult CML patients. The manuscript is very well written and the experimental workflow well designed. Despite that, I have some questions and concerns.
1. "Statistics". This paragraph does not report the Sample Size Calculation analysis. Please, report it in order to justify the robustness of the study.
2. The Authors report that the BCR::ABL1 transcript type doesn't affect the response considered as achievement of MMR within the first year of treatment. Did the Authors consider that RT-qPCR had been reported as amplifying b3a2 and b2a2 with different performance? Both Kjaer (PMID 30985947) and Bernardi (PMID 31233644) reported these results and both of them demonstrated that dPCR overcame this limit. Since the Authors of the present manuscript underlined the lack of accuracy of the conventional MRD assessment in a previous publication (PMID 30897165, Ref 33 of the present manuscript), confirmed by Zanaglio et al (PMID 33250471), did they considered to quantify BCR::ABL1 also by dPCR? Is the BCR::ABL1 transcript quantification by dPCR at the moment of MMR achievement possible? If the samples are not available or there are some technical impairments, I strongly suggest the Authors to add a comment about it in the "Discussion" section by citing the suggested papers.
3. Considering the lack of accuracy of RT-qPCR reported in the previous point, I suggest the Authors to perform a correlation between the ESPL1, PTTG1, and PTTG1IP expression levels and the slope of MRD expressed as IS% instead of MR classes. In fact, a minimal quantity of transcript (eg. 0.01% vs 0.0101%) may dramatically influences on the MR class determination. This analysis will be of pivotal importance in order to support the interpretation of the results.
4. The Authors reported the use of ABL1 as reference gene for IS% determination. Why did they change the reference gene (GUSB) for the relative quantification of the potential markers' transcripts?
I hope my comments/suggestions will help the authors in improving the quality of the manuscript and the robustness of the data.